# Solar Landscapes: A Methodology for the Adaptive Integration of Renewable Energy Production into Cultural Landscapes

**Chrili Car** *[ID], **Erwin Frohmann** [ID] and **Dagmar Grimm-Pretner** [ID]

Institute of Landscape Architecture, University of Natural Resources and Life Sciences, 1190 Vienna, Austria; dagmar.grimm-pretner@boku.ac.at (D.G.-P.)
* Correspondence: chrichrichri@gmx.at; Tel.: +43-1-47654-85200

**Abstract:** The increasing use of solar energy is an integral step toward carbon neutrality. At the same time, outdoor solar farms are significantly altering existing cultural landscapes. This work examines the possibilities of integrating the use of solar energy into these landscapes in such a way that the unique, regional character of places is preserved and enhanced. The research project that was carried out developed a conceptual design approach that takes as its starting point landscape architectural and aesthetic analyses of existing sites in Styria, Austria, the spatial characteristics of the cultural landscapes in which they are embedded, and their suitability for generating solar energy. The comparison of a site's characteristics with the technical possibilities evaluated from a literature review enables a responsive design practice using solar modules. The result is a method of landscape architectural design that integrates solar energy on the basis of an adaptive site-specific approach as well as a catalogue of sample cases that illustrate how designing with solar modules can honor and add value to existing places while enhancing their ecological, economic, and social functions.

**Keywords:** solar energy; photovoltaics; open space; research by design; cultural landscapes; energy landscapes; landscape aesthetics; Styria; Austria

## 1. Introduction

The current landscapes of Central Europe have been created by the interaction between natural conditions and human habitation over a period of 20,000 years since the time of the last Ice Age [1] (p. 87). Greenhouse gas emissions, caused by industrialization and the use of fossil fuels, have led to global climatic changes. In the Paris Agreement and the United Nations Framework Convention on Climate Change, 195 states and the European Union agreed to limit global warming to 1.5 degrees Celsius [2] (p. 22). The European Union's Green Deal envisages net zero greenhouse gas emissions by 2050 [3] (p. 2). In Austria, energy production was responsible for 43.8% of greenhouse gas emissions in 2018 [4] (pp. 5, 69). The national energy and climate strategy envisages a 100% transition to renewable electricity by 2030 [5] (p. 7). Austria's Renewables Expansion Act [6] (§ 4.4) targets an increase of 27 TWh in renewable energy production by 2030, 11 TWh of which are to be sourced from photovoltaics. This goal requires solar farms for the generation of 5.7 TWh of electricity to be built in open spaces, corresponding to an area of 91 km$^2$, based on the current state of technology, or 57 km$^2$, when forecasts for technical innovations in solar systems are taken into account [7] (p. 4).

Making the required space available means significantly altering existing natural or cultural landscapes, a matter which has been met with resistance by local residents. By contrast, the same group that opposes the building of wind and solar farms in their immediate vicinity often has a romantic image of traditional pre-industrial energy landscapes [8] (p. 14). The prevalent lack of acceptance represents a challenge to the expansion of the renewable energy sector [9] (p. 17). Raising the acceptance rate requires the involvement—largely absent to date—of the discipline of landscape architecture in designing and realizing solar energy systems. This has been identified as key to achieving the envisaged climate

goals [2] (p. 22) [10] (p. 64) [11] (p. 659). By developing multifunctional, inclusive designs [11] (pp. 629, 638, 650) and complementing building-integrated photovoltaics with landscape-integrated solar systems, landscape architecture can contribute to the further development of existing landscapes in a way that does justice to the legacy of the particular area of land at hand [11] (p. 659).

In the European context, Austria is a country with a diverse topography composed of 2600 different small-scale cultural and natural landscapes [12] (p. 9). The Biodiversity Strategy 2020+ [13] (p. 18) aims to preserve this diversity of flora, fauna, and cultural interaction as a natural and cultural asset and take regional characteristics into account when building wind or solar farms. Based on this, the Office of the Styrian Provincial Government of Austria commissioned the Institute of Landscape Architecture at the University of Natural Resources and Life Sciences in Vienna to develop, in 2022, a general design procedure and a catalogue that would exemplify site-specific design with solar modules.

To investigate the integration of solar energy production in alignment with the characteristics of the landscape, this study builds on the spatial discourse that investigates links between cultural landscapes and energy production. Jolanda de Jong and Sven Stremke [8] researched the historical transformation of landscapes in connection with the generation of energy during the last millennium in the Western Netherlands and studied the lessons to be learnt for the transition to renewable energy landscapes. Alessandra Scognamiglio [11] carried out design experiments to understand the impacts on agricultural fields of different layout patterns—created by variations in the density and orientation of standard crystalline solar modules—and how these patterns can relate to landscape characteristics and accommodate different functions. Simone Giostra [14] tested various arrangements of solar modules to create large-scale solar farms with standard modules based on parametric land analysis, taking into account potential functional synergies in four regions of Europe. Sven Stremke [15], meanwhile, has outlined different concepts, principles, and procedures that are involved in designing sustainable energy landscapes.

Taken together, these studies cover an array of topics that are relevant to the design of energy landscapes. They take the form of individual case studies or analyze design approaches while focusing on standard mono- or polycrystalline solar modules, and they have not developed a general procedure for landscape designers or spatial planners that ensures that the design process reflects the site's characteristics nor have they considered the full spectrum of structurally different solar modules that are available. The study this paper is based on synthesizes the different aspects—the historical development of landscapes, the arrangement of design interventions, synergies with other relevant landscape functions, and principles of designing with solar modules—while filling in knowledge gaps: e.g., by compiling a systematic catalogue of the different types of solar modules, which can serve designers as a toolkit for the integration of solar modules into existing cultural landscapes. This study investigates, furthermore, a general procedure for the adaptive integration of solar energy—an adaptive integrative design approach in the sense that the solar infrastructure is tailored to the places where it is developed and reflects the site characteristics so that it preserves or enhances the unique regional character of its location, both functionally and aesthetically [11]. This research thus acknowledges that places are part of connective landscapes, which are unique, offer a variety of functions, and have evolved as a result of geological processes and the co-habitation of flora and fauna on the land [12] (p. 9). If these landscapes have been significantly influenced by human co-habitation within the site-specific natural conditions, they are regarded as cultural landscapes.

In investigating this general procedure and showcasing exemplary bespoke design solutions that have been developed on the basis of it, the goal is to enrich, foster, and provide inspiration for the transdisciplinary discourse between decision makers, including political representatives, spatial planners, landscape architects, and engineers. This research sets out to contribute to a more sustainable practice and establish a framework that fosters interdisciplinary project development, distinguishing it from current practice, which rarely

has adequate spatial planning frameworks in place and often does not include the expertise of disciplines such as landscape architecture [11] (p. 659).

## 2. Materials and Methods: Research by Design

To develop a general procedure for the site-specific integration of solar energy generation, this research project follows the approach of research by design [16] (pp. 221–222). Research by design investigates design case(s) and derives general insights from design experiments through a discussion of case-specific design processes in the context of the wider discourse, following an overall research design which makes it possible to produce a general result. To integrate solar energy usage in a sensitive way into different cultural landscapes and create synergies between the overall functional demands, an inductive procedure is chosen that includes the following methods (Figure 1):

1.  The investigation starts with *site analyses* based on the selection of suitable exemplary locations of a reference landscape (see Section 2.1).
2.  A *literature review* generates an understanding of the current state of the technology, the existing possibilities, and innovative trends in the development and application of solar modules. A typology is created as a toolkit necessary for subsequent design experiments (see Section 2.2).
3.  Conceptual design proposals are developed for the integration of solar energy generation in the selected places through a landscape architectural design analysis (see Section 2.3).
4.  The design experience and procedure are reflected upon through a review and discussion of the relevant literature sources (see Section 2.2), which leads to the formulation of a design method as a step-by-step procedure.

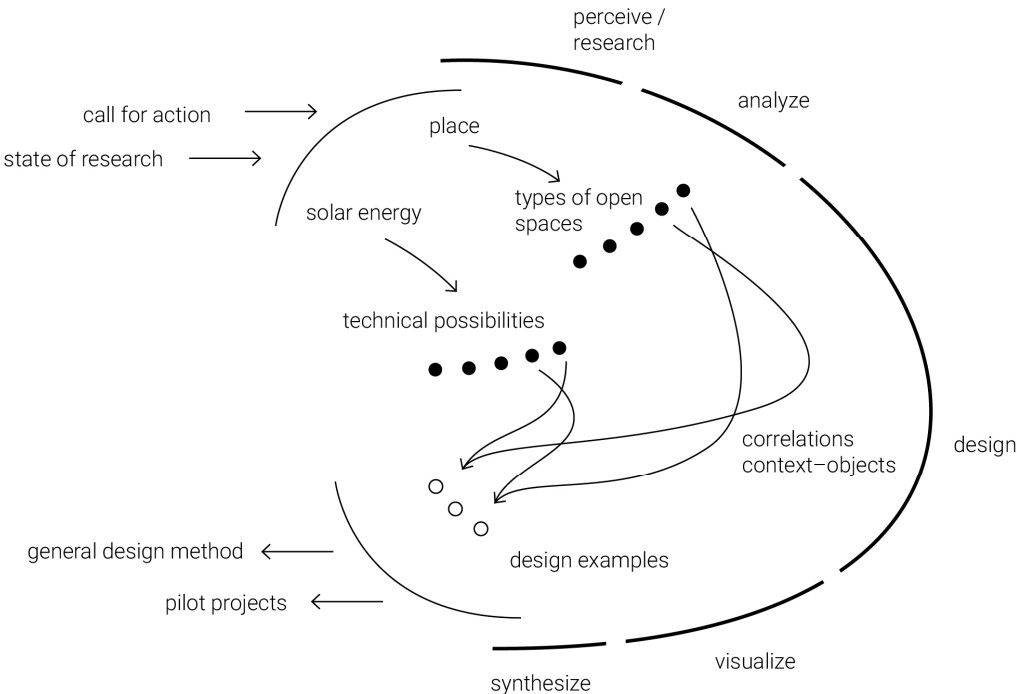

**Figure 1.** The research-by-design approach adopted in this study.

### 2.1. Landscape Architectural Site Analysis

In the first step, phenomena are observed at specific sites, and their frequency of occurrence is tracked. A semi-structural analysis based on walks-as-dérives investigates the site's spatial configuration, characteristic atmospheric elements, and functional qualities, culminating in a selection of places for further study [17] (pp. 1–2) [18] [19] (p. 180). Background information, aerial views, photos, and descriptive landscape portraits serve to

assign the open spaces to general types and make their spatial connectivity comprehensible as characteristic landscapes. Furthermore, an evaluation is made of whether solar energy can be suitably integrated using criteria from a spatial planning perspective [20] (§ 3):

- Landscape and settlement areas with strong cultural influences: highly suitable, priority areas;
- Landscape and settlement areas with some cultural influences: moderately suitable, sensitive interventions;
- Pristine natural and cultural heritage landscapes: suitable only in exceptional cases, for small-scale interventions t particular consideration to the unique landscape features.

The Leibnitz district in southern Styria, Austria, was selected as a reference landscape for the purpose of investigating the integration of photovoltaics and solar thermal energy into rural open spaces (Figures 2 and 3). This district is characterized by its small-scale, diverse rural composition [21,22] (p. 83), and its location on the transnational traffic axis between Graz and Maribor, characterized by large-scale industrial and commercial zones. The district's diversity—in terms of both its functions and natural conditions—allows for a broad spectrum of design cases to be tested.

### 2.2. Literature Review

The site analysis in this work is paired with a literature review [23] (pp. 6–8, 11) of material options and development trends. An interpretive evaluation of selected literature sources reveals the current state of knowledge with regard to the technical possibilities for generating solar energy for the production of electricity or heat. This investigation includes scientific papers, reference projects, product catalogues, and data sheets, and allow a material catalogue to be created which also considers examples of the application of such possibilities in design projects as well as their state of development and their energy yield.

In addition, the literature review facilitates an understanding of the current spatial discourse and the criteria used in selecting suitable sites; at the same time, it provides this study with a methodological basis for formulating a general design procedure.

### 2.3. Landscape Architectural Design Analysis

Based on information about the sites' characteristics and the material catalogue, an experimental concept design is developed for the specific places by means of a landscape architectural design analysis, which investigates the structural, functional, and aesthetic correlation of a site not only with its context but also with suitable solar design elements [24] (p. 23). This analysis considers synergies with other relevant functions onsite, so that the integration of solar energy amplifies the existing landscape characteristics by taking into account ecological, social, and economic requirements. The technical solar modules are treated in the design process in the same way as all the other landscape architectural elements that are relevant for the coherent and historically grown landscape system of the selected places. The outcome of the design analysis is a pool of ideas that provide visionary solutions for the selected places, with the intention to showcase and provide inspiration on how the integration of solar modules in the design of open spaces can relate to their specific characteristics. In addition, the design experience of developing a series of designs for a variety of different open spaces contributes to the formulation of a general step-by-step procedure that ensures, case-by-case, that the integration of solar energy modules relates to the site-specific characteristics of cultural landscapes.

## 3. Results

### 3.1. Suitable Sites for the Integration of Solar Energy

The site visits led to the selection of twenty-three open spaces (Figures 2 and 3) that were assigned by typology to six categories according to the following functions: settlements, commerce, traffic, leisure, agriculture, and infrastructure. The selection includes the open spaces of schools, business parks, town squares, different types of infrastructure,

sports grounds, parks, agricultural spaces, and sites with various natural conditions, such as a hilly topography, flat land, and water bodies.

In order to create a catalogue of different spaces for the integration of solar energy generation, each of the sites was portrayed using a photo, a zoning plan with areas eventually to be selected for the integration of solar energy, a site-specific landscape description, and a classification of its suitability that defines the recommended intensity of the intervention (Figure 4).

Twenty-one of the twenty-three selected open spaces were identified as suitable, at different intensities. The two open agricultural areas—a hop field and a vineyard—indicated the necessity of restricted zones. Natural landscapes—open, exposed sites in particular—were regarded as sensitive protection zones in which no or only small-scale installations were recommended as landmarks with a sculptural effect, while open spaces in the settlement area, the transition area to the open cultural landscape, or the environment of infrastructure systems were deemed particularly suitable for the integration of solar energy generation, such as spaces in and around existing villages and towns, commercial areas, and areas altered by a previous cultural impact or areas with low sensitivity outside the settlement area, such as existing infrastructure sites [20] (pp. 12, 22).

### 3.2. Possibilities of Designing with Solar Energy Modules: A Material Catalogue

The selection of suitable solar modules and their design effect is a key element in developing unique, site-specific solutions. The literature review revealed seventeen types of solar modules (Figure 5), which were collected in a material catalogue with a data sheet for each of the types which includes standard sizes, average energy efficiency, load-bearing capacities, and the state of development.

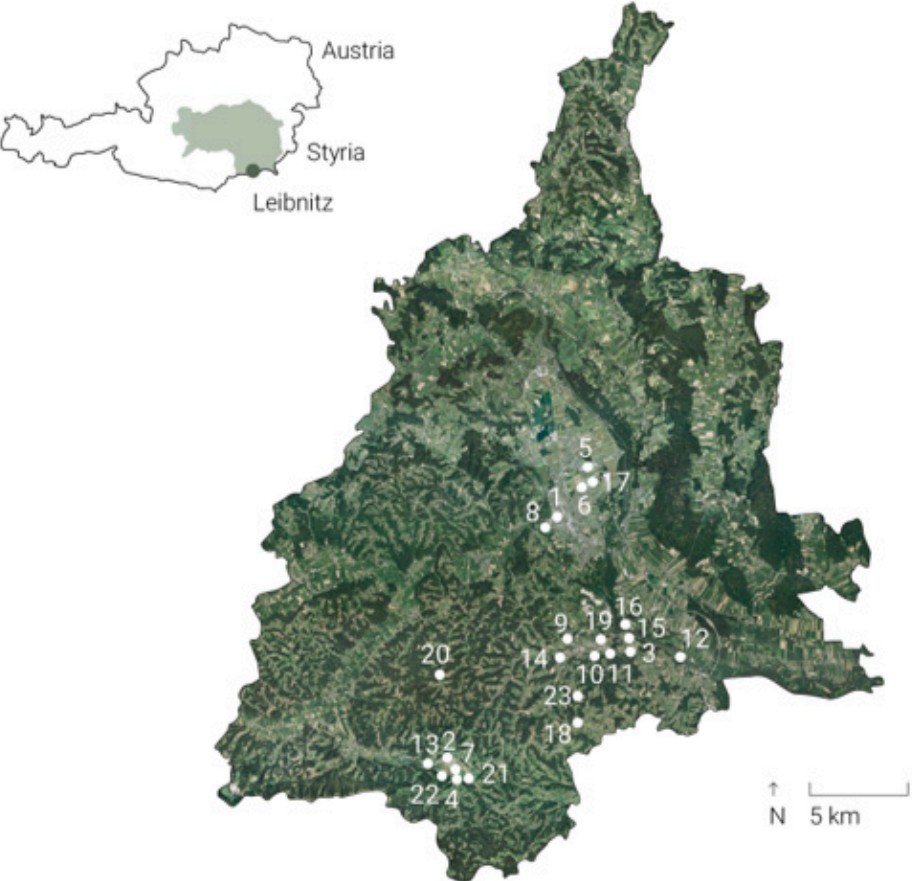

**Figure 2.** Location of the 23 sites (see Figure 3) that were studied in the reference landscape of the Leibnitz district in Styria, Austria; geodata from [21].

a Parking Lot  b Center with Advertising Tower  c Access Roads with Billboards

1 Main Square Leibnitz

2 School Campus Leutschach

3 Wine Store Erzherzog Johann

4 Business Park Leutschach

5 Business Park Leibnitz I

6 Business Park Leibnitz II

7 Outdoor Pool Leutschach

8 Sports Grounds & Pool Leibnitz

9 Sports Grounds Gamlitz

10 Motor Skills Park Gamlitz

11 Flood Protection Dam Gamlitz

12 Mur Dam

13 Sewage Treatment Plant Leutschach

14 Wood Chip Mill Gamlitz

15 Ehrenhausen Railway Station

16 Mur Bridge Ehrenhausen

17 Roundabout

18 Rest Area Watschgerei

19 Observation Tower Gamlitz

20 Lookout Kreuzbergwarte

21 Private Garden

22 Hop field

23 Vineyard

Additional spaces can include: e.g. Highway Noise Barriers and Cloverleaves, Brownfields and Vacant Lots, Farms and Homesteads

**Figure 3.** Selected places for investigating suitable ways of integrating solar energy generation; geodata from [21].

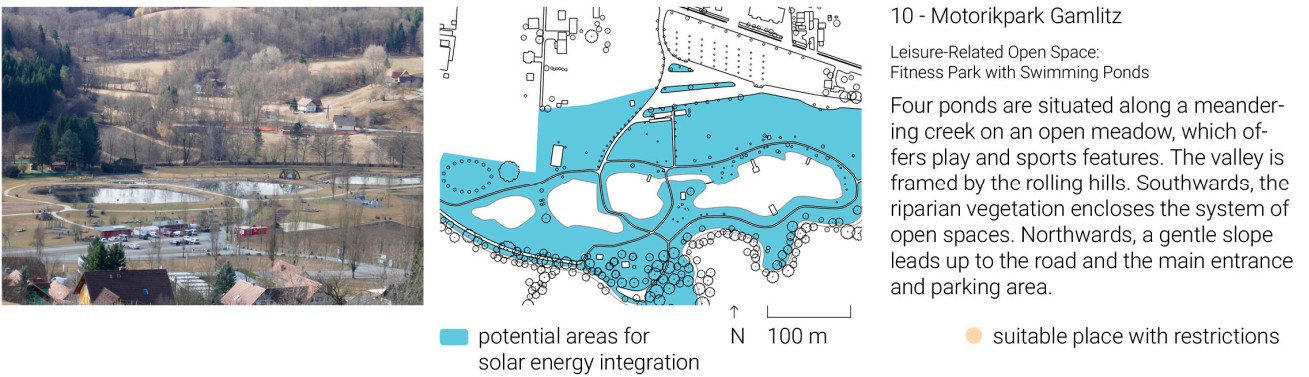

10 - Motorikpark Gamlitz

Leisure-Related Open Space: Fitness Park with Swimming Ponds

Four ponds are situated along a meandering creek on an open meadow, which offers play and sports features. The valley is framed by the rolling hills. Southwards, the riparian vegetation encloses the system of open spaces. Northwards, a gentle slope leads up to the road and the main entrance and parking area.

■ potential areas for solar energy integration     ↑ N   100 m     ● suitable place with restrictions

**Figure 4.** Example of the site portrait of Motorikpark Gamlitz with a photograph, site plan, landscape description, potential areas of intervention, and assessment of its suitability for the integration of a system for solar energy generation.

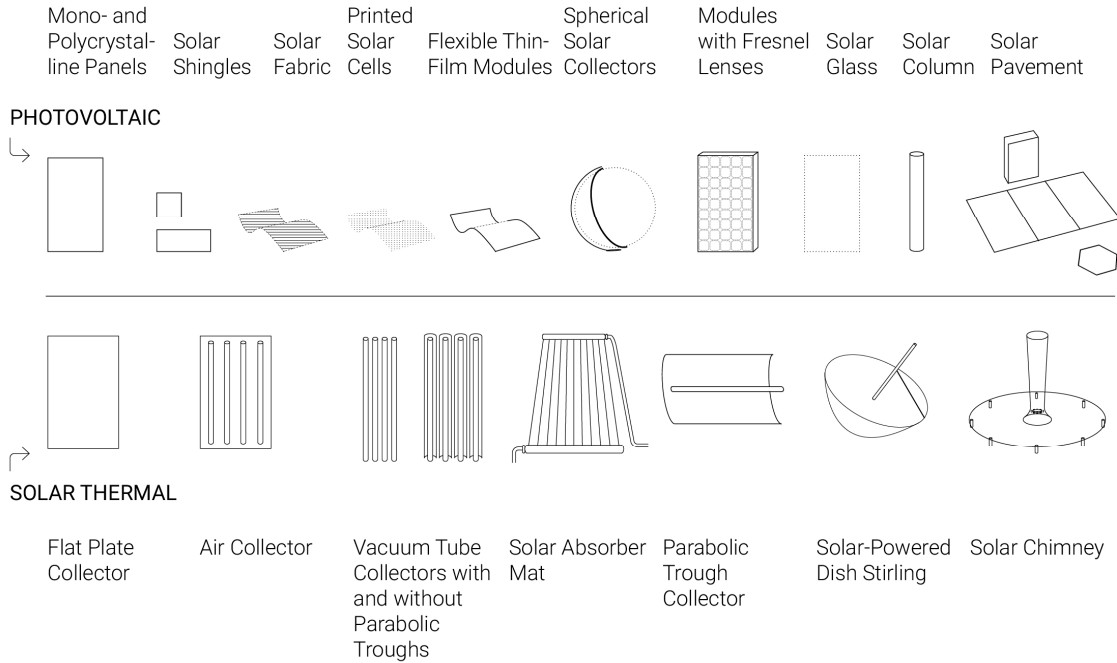

**Figure 5.** Overview of the different structural types of photovoltaic and solar thermal modules as part of the developed design material catalogue; data from [25–38].

The catalogue of sample solar modules depicts the current state of technological development. Included in the material catalogue are both established mass-produced solar modules and recent innovations that are in an experimental stage of development. By incorporating ongoing innovations in the field of solar energy modules, the catalogue is an open collection that provides designers with a toolkit and makes them aware of existing possibilities for designing with solar modules (see Section 4.4): the options include solar shingles, flexible thin-film cells, and spherical solar lenses as a wide variety of types of solar modules are available as design elements [39] (pp. 10–13, 22), [34,40] that have different qualities in use. Their technical suitability needs to be considered in the selection process: for example, ground-mounted solar modules as pavers for roads or pathways require heavy-duty, anti-slip, and scratch- and impact-resistant glass covers. Flexible thin-film modules can be suitable for applications on curved surfaces and tensile structures with low load-bearing requirements.

In addition to their different technical qualities, there are possibilities for adapting the look and feel of the modules, which enables the modules to be customized to suit the site's characteristics by coloring the cells, coating the surfaces, dying the protective cover glass, photo-laminating the cover glass, or using a relief cover glass [41] (pp. 1–3) [42–48]. These customizations go hand in hand with reduced efficiency, varying from a minimum of 2.5% for dyed solar cells to up to 40% for photo-laminated cover glass [42,44]. At the same time, adapting the appearance enables solar modules to be adjusted to the local color scheme or texture and avoid irritating reflections of sunlight, which is especially relevant for the application of these units in open spaces, while enabling functional synergies (e.g., photo-laminated solar modules as information or advertising boards).

### 3.3. Integrative Design with Solar Energy Modules: A Pool of Design Ideas

Based on the site examples which were studied and the material catalogue, ten concept design ideas were developed and visualized, representing a broad spectrum of different types of open spaces in the reference landscape (Figure 6). The pool of ideas that was developed illustrates how the design of solar energy generation systems can preserve established cultural landscapes and develop these landscapes further, in an inclusive way,

through a design which refers to the special characteristics of the landscape and thereby creates site-specific solutions [11] (pp. 65, 629, 638).

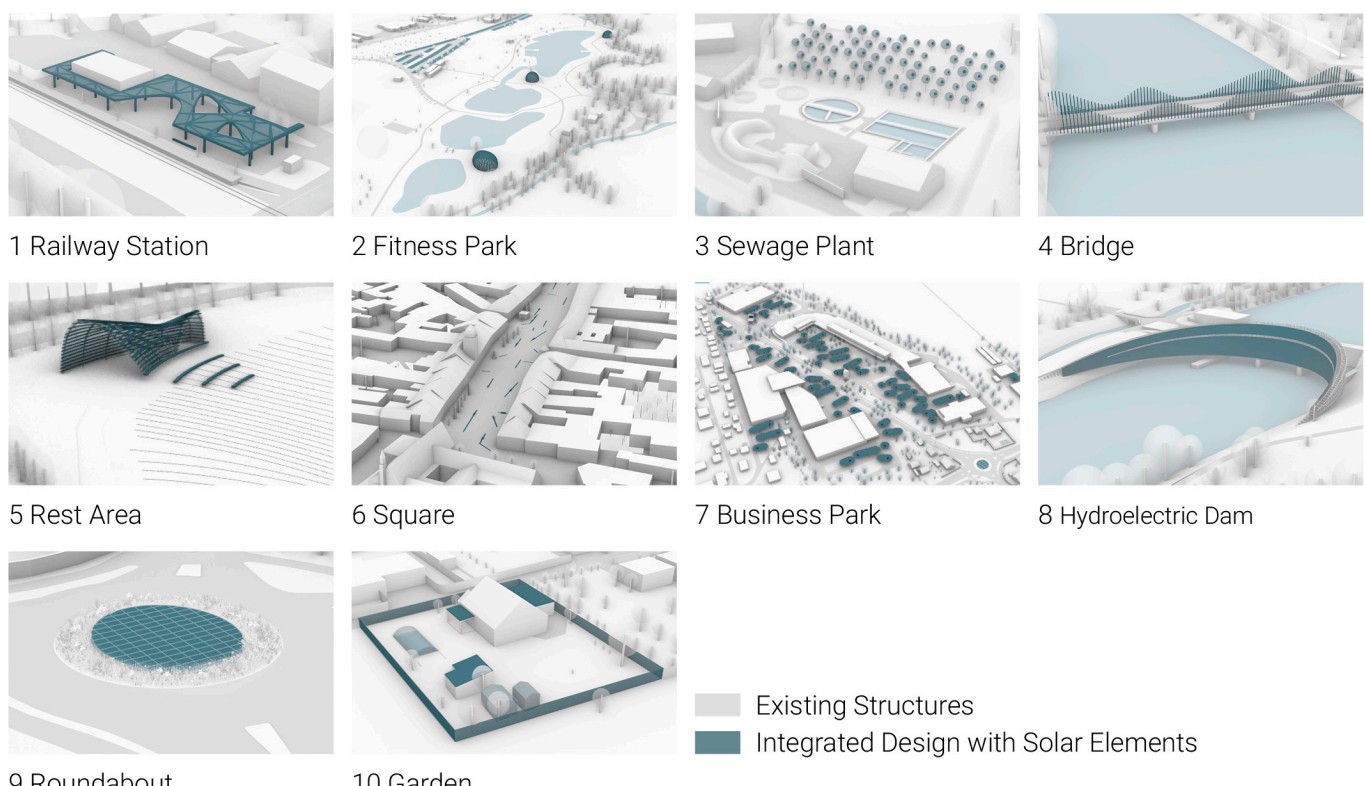

**Figure 6.** Overview of the design studies developed as part of the pool of ideas.

A lively topography creates visual axes with views of distant landscape features. Vague spatial boundaries benefit from elements of spatial order. An integrative design with solar modules accentuates existing spatial characteristics, satisfies the demand for missing functionality, and evokes associations related to the landscape in which the places are embedded, thereby promoting the unique local character of the cultural landscape (Figure 7). This was achieved, in the design process, by selecting core site characteristics from the atmospheric landscape portraits as the basis for a structural design mode. In the next step, solar elements and suitable colors and textures were chosen that corresponded to the design concept. The different functions of the place were then considered, and potential surplus value was assessed and integrated into the design with the selected solar elements.

The resulting concept designs include solar domes for physical exercise that become a series of landmarks between a system of existing ponds; a celestial eye with integrated beekeeping in a roundabout; and a solar kiosk as a landscape stage that emerges from the existing topography. The developed pool of ideas considers all parts of the solar energy intervention at hand, including the support structure, which, in some cases, becomes a habitat for animal and plant species or a shelter for hikers. Such functional synergies take into account existing demands on the open space and also create new possible uses for additional ecological, economic, and social value [49] (p. 87).

The ideas were developed in response to generalized types of open spaces—such as train stations, roundabouts, and sewage treatment plants—which recur in different locations. But the landscape and site configuration in which these types of open spaces are located are unique. Thus, the design examples were not developed in order to be replicated 1:1 in a repetitive modular way at different locations. Instead, to reinforce the characteristics of unique places, new site-specific solutions are to be developed each time. To provide a general guide for this, a method was developed based on this design experience.

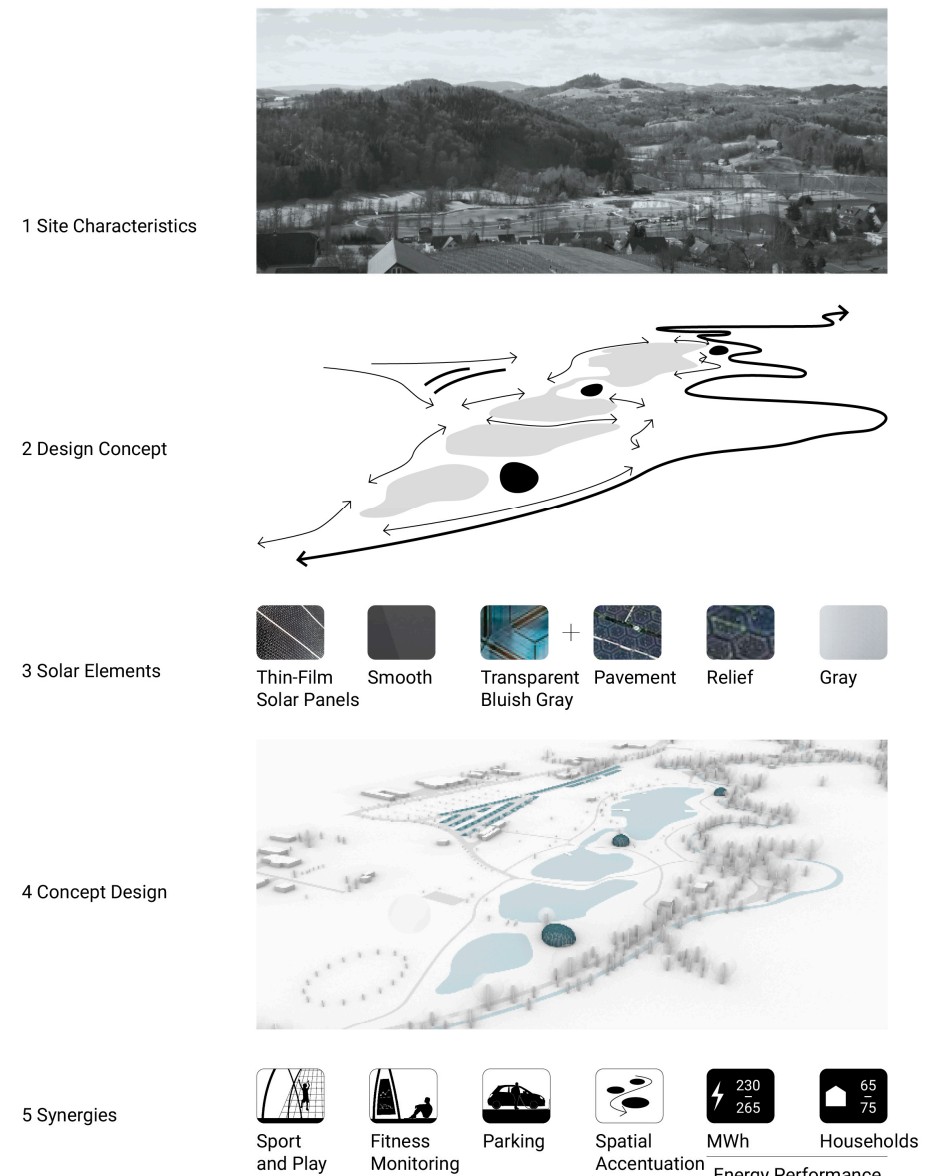

**Figure 7.** Concept design development based on the example of Motorikpark Gamlitz: (**1**) Four existing, dynamically shaped ponds are situated along a meandering creek at the bottom of a valley, whose gentle slopes gradually become steeper and culminate in the contours of rolling hills. (**2**) Three monumental, symbolic "river pebbles" placed at focal points in the space rhythmically accentuate the waterbodies: centrally, at the solar plexus of the pond sequence, and within bays and creek meanders. Additionally, lines along the site's physical contours highlight the topography that leads down to the ponds. (**3**) Smooth thin-film modules with a bluish-gray tone match the color of the ponds. The linear pavement of the parking spaces along the slope consists of solar road panels with a size of 0.7 × 1.26 m and protected by resilient anti-slip gray relief glass. (**4**) The three pebbles consist of a grid shell, on which flexible thin-film solar panels are attached on tension wires. (**5**) The three pebbles feature different motor skills programs: climbing nets, a labyrinth, and gymnastics facilities with digital fitness monitoring. The solar pavement upgrades the existing unsealed parking spaces. The photovoltaic modules generate an output of 230 to 265 MWh and can supply sixty-five to seventy-five households with electricity.

### 3.4. A Methodology of Integrated Landscape Architectural Design with Solar Energy Modules

A five-step design procedure supports a landscape-sensitive and site-specific integration of the solar elements based on the design experience of the test cases by applying the above catalogue of solar modules (Figure 8):

0.  Precondition Site Suitability and Selection: The open space is available for the design of energy landscapes and suitable for the integration of solar modules. The integration of solar power is compatible with the existing functions of the spatial context. Energy production is relevant on-site. Places influenced by previous cultural impact are suitable in most cases, such as in or near settlements or infrastructure. Places characterized by pristine nature and ecology, places of special cultural heritage, and open, exposed sites are unsuitable.

1.  Analysis of Site Characteristics: The spatial, social, ecological, and economic values of the selected place are analyzed, along with its interconnectivity with the region. Unique features that make the place special are identified as well as the characteristics of the region that are prevalent on-site or—if absent—could be reinforced through design.

2.  Design Concept: A design mode is developed that reflects and enhances the special characteristics of the site or region while taking ecological, social, and economic requirements into account. This step defines the type and intensity of the intervention, its patterns, and structure as well as the atmosphere that is to be created. This design mode targets—but is not limited to—the integration of solar energy generation and also considers the full spectrum of landscape architectural design features such as vegetation, circulation, lighting, and social features for all generations, including meeting places, rest areas, and play features, to the extent that they are locally relevant.

3.  Selection of Suitable Solar Elements: According to the site-specific design approach, types of solar modules are selected, and their material, color quality, and surface finish are defined. Solutions are chosen that fit the site in terms not only of their technical suitability and durability but also of their resonance with and enhancement of unique spatial site conditions and synergies with other relevant functions on-site. This step takes place with an eye to the specific project's potential for innovation and the possibilities for experimenting with new techniques.

4.  Concept Design: Design plans are drafted on the basis of the developed design approach and the selection of design elements, taking into consideration the full spectrum of landscape architectural design features, with solar elements as one aspect of them.

5.  Evaluation of the Overall Spatial Effect and Value Engineering: In addition to generating energy, the design proposal adds locally relevant social, ecological, and economic value and strengthens the spatial identity of the location. The design concept is reviewed and refined by considering the existing and potential enhancement of the multifunctionality of the selected place, optimized to promote synergies in response to the spatial, social, ecological, and economic demands.

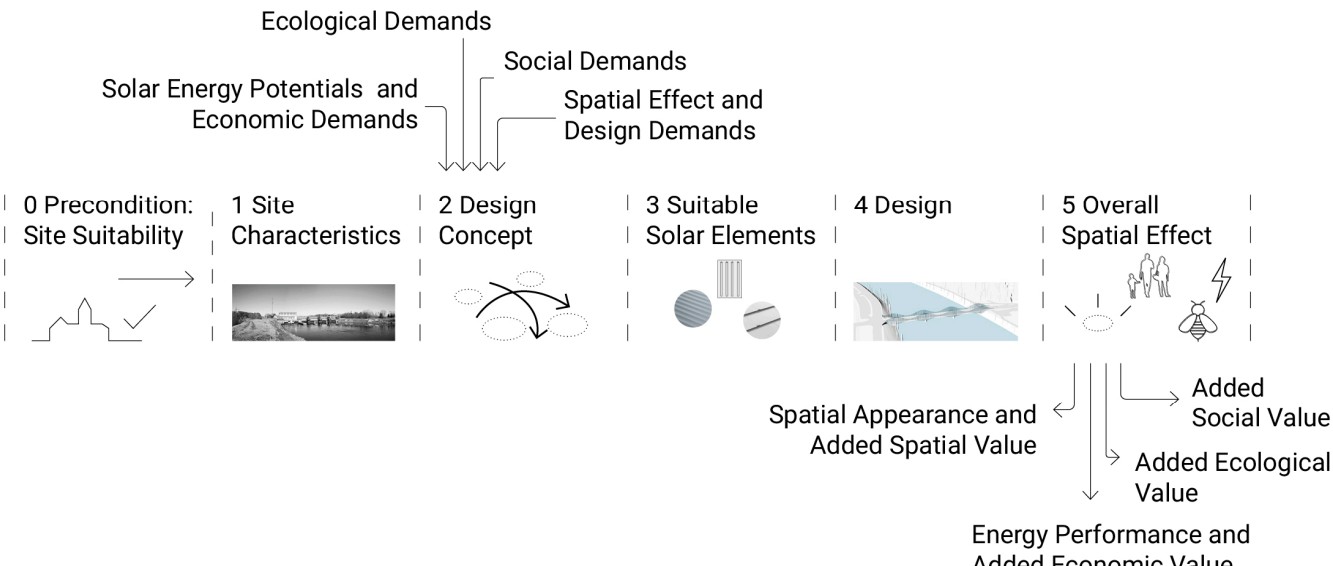

**Figure 8.** Steps for creating landscape architectural designs that integrate solar energy.

The developed method is a further evolution of the standard method of landscape architectural design tailored to meet the specific requirements of integrating solar energy in open spaces. The outcomes of this method in practice are conceptual design solutions that embed solar energy generation in existing cultural landscapes by accentuating the genius loci and generating functional synergies.

## 4. Discussion

### *4.1. Energy and Landscape*

Cultural landscapes are the result of the interaction between people and their natural environment, in which cycles of energy flows play an essential role. The cultural interaction involved in making these natural energy flows available to benefit humans shapes ecosystems as much as natural factors shape human systems and lifestyles [14] (pp. 2, 22) [50] (p. 4808). Natural solar energy landscapes exist on the basis of plant photosynthesis, which produces energy-rich biomolecules from lower-energy substances using sunlight. This process determines the habitus and growth behavior of plants and, therefore, has a direct influence on their spatial structure. The transformation of energy is an essential component that is integrated into the entire plant organism or plant community: solar receptors take the shape of the leaves and reflect other functions aside from energy production, such as, for example, the structural qualities and ability to transport molecules in green stems and their vascular bundles [51] (pp. 255–263).

In traditional cultural energy landscapes, water mills and windmills constitute landmarks that contribute to the identity of the landscape. By contrast, industrial energy landscapes have—on a large scale—singular functions separated from other uses. These monofunctional energy landscapes, thus, tend to be alien to the cultural diversity that has evolved over a long period of time through the interaction of natural processes and human activity on-site [8] (p. 14). The developed method and design concepts combine the adaptive way in which energy production is integrated in the cells for photosynthesis in plant organisms with the multifunctional character of traditional energy landscapes, where the generation of renewable energy takes place as one function among many.

### *4.2. Smart City, Smart Country*

The developed method ensures functional synergies instead of focusing on a maximum energy output that replaces other uses, similar to the Smart City approach, which reflects traditional spatial multifunctionality and attempts to reunite the functions which have become increasingly separated in line with the concept of industrial modernism. Infrastructure, transport, energy supply, education, and healthcare become part of the physical urban space and are made available with the help of digital information and communications technology [52] (pp. 1, 8) [53] (pp. 2–4). The integration is comparable to metabolic processes in natural landscapes. Similar to photosynthesis receptors as part of holistic plant organisms [51] (pp. 255–263), intelligent modules, surfaces, and sensors are integrated into built features, vehicles, and everyday appliances. The city is seen as an overall system in which the digital meta-level simplifies all areas of daily life in a user-friendly manner.

There is no justification for restricting this approach to urban areas. The Smart Country is to be understood as a complement to the Smart City concept by providing the same opportunities to wider regions through the hybrid integration of additional functions into both urban and rural spaces [49] (p. 87). This step fosters a decentralized independence of supply and is particularly important for rural areas [53] (pp. 1, 6, 7) [54] (p. 14). A decentralized energy supply through intelligent surfaces, as promoted by the developed method, contributes to Smart Countries, which have all functions—including ecology—integrated and, thereby, foster the attractiveness of rural regions as places to live and work, which balances the current inequality between cities and countryside [55] (pp. 77) [56] (p. 26).

### *4.3. Transcending the "Not in My Backyard" Mentality*

Lack of reference to the local context results in the prevalent "not-in-my-backyard" attitude to renewable energy production among sections of the population [9] (p. 17) [10] (p. 55), who profess a general awareness of the necessity to promote the renewable energy sector and favor its development while rejecting any actual realization of it within their own personal sphere. Solar systems that consider the distinctive character of landscapes through landscape design [10] (p. 64) do not represent an alien element when designed using the method that has been developed; rather, they elaborate appropriately on the 20,000-year-old history of the landscape [1] (p. 87). The inclusive, integrated design of solar landscapes formally corresponds to unique landscape features and increases the functional diversity of landscape areas, e.g., by contributing to their ecological mosaic and niche structure [11] (pp. 629, 638, 650). Landscape features with a human scale and references to unique elements in the landscape are essential if wider acceptance is to be achieved, as is an inclusive design which refers to the landscape characteristics and the ecological, social, and economic value on-site and, thereby, creates unique, site-specific solutions.

### *4.4. Innovation*

In terms of application, the developed method supports site-specific design solutions and enhances the potential of existing uses. Adaptive solutions for the integration of solar energy in open spaces require versatile solar modules that can be integrated in an adaptive way. Current technical research is leading to a boost in development in the performance, cost-effectiveness, and diversification of the types and appearance of solar modules [10] (pp. 2, 7, 45, 64). Furthermore, increasing efficiency is making the energy modules less dependent on direct orientation to the sun. This trend in technical development is increasingly leading toward flexible solar modules in a variety of shapes as smart surfaces that can be flexibly applied to any objects in open spaces or buildings, such as small-sized solar shingles, flexible thin-film cells, solar fabric, and printed solar cells. This tendency is to be seen as a task for future technical research to enable the development of more adaptable solar modules that allow photovoltaics and solar thermal energy to be seamlessly integrated into the design elements in open spaces and can, thereby, contribute to the unique character of cultural landscapes.

### *4.5. Integration of Solar Energy as a Design Element*

The design of places such that individual design elements are coordinated with the regional context follows the standard landscape architectural design process [24] (p. 23). Each design element has its spatial requirements and opportunities. The developed method works with solar modules as one type among a range of elements of the repertoire of landscape architecture. While designing with solar modules takes place, in principle, in the same way as designing with other landscape design elements, the method is especially geared to introducing energy generation as additional value in a unique, coherent landscape. The general method of landscape design analysis has been refined with regard to renewable energy generation, with clearly defined criteria on how suitable places are selected and spatial characteristics are translated into the design of solar modules in an integrative way—much like the solar receptors for photosynthesis are integrated into the habitus of plants and traditional energy landscapes are directly tailored to the multifunctional landscape context [8] (p. 14) [52] (pp. 255–263)—not by taking existing designs and copying them elsewhere but by designing a bespoke solution case-by-case in response to the unique situation at hand. Landscapes become solar landscapes not by replacing existing characteristics and uses but by reflecting on and enhancing them.

The developed landscape architectural method establishes the profession of landscape architecture as an integral player in the conversation about the transition toward renewable energy—one of many disciplines which play all a relevant part in further investigating strategies for fostering the use of solar energy. In order for sensitive designs involving photovoltaic and solar thermal systems to become mainstream, it is necessary for spatial

planning instruments to be coordinated: incorporating the design of renewable energy systems into regulatory frameworks should include criteria for the selection of suitable places, a design approach that is sensitive to the landscape and site-specific characteristics, the establishment of interdisciplinary advisory boards, measures for interdisciplinary participation, the creation of awareness among the general population, and a diverse range of available technical solutions as well as targeted incentives and taxation.

**5. Conclusions**

The investigation of an adaptive approach to the integration of solar energy generation showed that energy production is not necessarily antithetical to the preservation and further development of evolved cultural landscapes, whose characteristics are respected while multiple uses are catered to. Used as a framework for site-specific design, the method that was developed in this paper, therefore, offers a general approach that leads to bespoke spatial responses to enable the transition to renewable energies while taking into account the characteristics of cultural landscapes—it does not provide a set of solutions which can be applied across the board. Integrating the use of solar energy into these cultural landscapes has the potential to create Smart Countries, promote decentralized organization in rural environments, and, thus, balance urban–rural disparities.

In practice, this method fosters adaptive site-specific solutions and functional synergies. This requires a diversification of the applied solar modules in mainstream applications and, thus, supports technical innovation to further develop adaptive modules. While the method introduces landscape architecture as a key discipline in transforming the energy sector, this research acknowledges the importance of further integrating designs that use renewable energy into the frameworks of spatial planning and the necessity of incorporating technical and engineering innovations in adaptive solar modules.

**Author Contributions:** Conceptualization, methodology, investigation, and validation, C.C., E.F. and D.G.-P.; resources, E.F. and C.C.; formal analysis, visualization, data curation, and writing—original draft preparation, C.C.; writing—review and editing, C.C., E.F. and D.G.-P.; project supervision and administration, E.F. and D.G.-P.; funding acquisition, E.F. All authors have read and agreed to the published version of the manuscript.

**Funding:** This research was funded by the Government of Styria, Department 15: Energy, Housing, Technology, funding number ABT 15-230/2023-42.

**Institutional Review Board Statement:** Not applicable. No humans, other than the authors, or animals were directly involved in this study.

**Informed Consent Statement:** Informed consent was obtained from all the subjects and institutions from which research material was requested.

**Data Availability Statement:** The data presented in this study are available on request from the corresponding author. The data are not publicly available. The data have not been released for general publication by the funding institution as follow-up studies are currently under development.

**Conflicts of Interest:** The authors declare no conflicts of interest. The funders had no influence on the design of the study; on the collection, analyses, or interpretation of the data; on the writing of the manuscript; or on the decision to publish the results.

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
