# Peer review of "Solar Landscapes: A Methodology for the Adaptive Integration of Renewable Energy Production into Cultural Landscapes"

_sustainability, doi:10.3390/su16052216_

Round 1
Reviewer 1 Report
Comments and Suggestions for Authors
This study is kind of interesting. Some issues are listed below:
1. The contributions are not clear. Please present them in the Introduction.
2. The formatting is not well organised. Figure 2 covers some words above it. The last paragraph is not start with blank space.
3. The Materials and Methods are not well presented. Please give a structured section.
Comments on the Quality of English LanguageQuality of English Language is fine.
Author Response
Thank you for your feedback. We revised the paper and considered your comments in the following way:
ad 1. We added an explanation of basic terms, stated the existing discourse with key literature that served as the starting point of our research, and in which way the research project contributes to the discourse.
ad 2. We reviewed the formatting and ensured images are embedded correctly.
ad 3. We restructured the chapter methods with a general introduction and the different applied methods as sub-chapters.
Reviewer 2 Report
Comments and Suggestions for Authors
Authors have outlined the utilization of landscape architecture to better accommodate the solar module installation projects within a cultural landscape. Though the method is focused for the region of Styria, Austria but general enough to be utilized in other similar locations.
Article is well written and the design procedure outlined here might be useful for relent community of designers.
Following are the suggestions to make the article better:
1. In section:3.4, authors have outlined the design methodology for the integration of a solar module within an existing cultural landscape. In the 'selection of suitable solar module' special consideration is given such that site characteristics does not get altered significantly by the solar modules. What about the solar modules getting effected by the site characteristic? E.g., proximity to a water body might not be ideal location for durability of certain solar cells.
2. Is there any regulation related things that need to keep in mind? Installation of a standalone solar modules might reduce the effective open space of a region e.g., green parks.
3. In section:3.2, authors have outlined several solar module structures available. A brief description on what kind of solar modules might be ideal for a site based on it's characteristic with some example is desirable.
4. Is there any specific site-characteristic that is not suitable for the installation of solar modules and hence those regions should be avoided all together for such projects?
Author Response
Thank you for your feedback. We revised the paper and considered your comments in the following way:
ad 1. Solar Farms that float on water bodies, or solar-powered fountains with the solar cells under water already exist. With the ongoing innovations in the field of solar modules, the research project does not limit the application of certain types of modules to certain site characteristics, but starts with investigating sites and their characteristics and chooses solutions that fit in a second step - in terms of technical suitability, and in terms of solutions that reflect and enhance the unique spatial, as well as that allow for synergies between different relevant functions on site. We added a more detailed explanation of the selection process that ensures the use of suitable modules to section 3.4. (3.).
ad 2. The consideration of synergies between the different relevant functions on site is an integral part of the developed method. Step 5 of the design with solar modules (section 3.4. (5.)) evaluates and optimizes the design with respect to other uses on site. This ensures e.g. that a standalone module (if chosen as suitable in step 3) would not limit any use in a green park, but become e.g. a shelter as a resting or meeting place.
ad 3. We added a statement and examples on the compliance and qualities of solar modules to section 3.2.
ad 4. There are areas that should be avoided altogether: Places characterized by pristine nature and ecology, as well as places of special cultural heritage are unsuitable. We outlined this in reference to spatial planning frameworks in section 3.4. (0.) and 2.)
We furthermore re-structured the section materials and methods and outlined the contribution of the research project to the existing discourse in the introduction.
Reviewer 3 Report
Comments and Suggestions for Authors
The manuscript announces the contribution with the methodology to integrate solar panels (as devices to enable source of renewable energy) into different spaces, having their design considered.
What was particularly emphasized in this paper:
1) Methodology
2) Adaptation
3) Adaptive integration
4) Cultural Landscape.
Authors should change paper:
1) The paper provides general sentences and no clear contribution with a methodology to enable integration of solar panels into particular types of spaces. The whole paper and the proposed methodology present well-known and obvious statements and suggestions. No clear and precise methodology, that would be innovative, comparing to common-reasoning related to this subject.
2) It would be beneficial to have Background and Related work sections after the introduction. Background should explain basic terms such as adaptive integration and cultural landscape. Related work should provide more details about previously published papers regarding the concept of adaptive or adaptable spaces and integration of different renewable energy devices into spaces having different utilization.
3) The proposed methodology should be presented via diagram, so the input and output of the proposed process of integrating solar panels with different places could be more understandable.
4) Adaptive systems are self-regulatory and they adapt according to surrounding events. In the context of integrating different places with renewable energy devices, adaptivity could be considered only if it is automated = adaptive solution to the changing environment. This could include using sensors to detect changes, artificial intelligence solutions for smart data processing and effectors to perform the change. Currently, this paper does mention smart city and smart country as concepts, but the idea is not elaborated with appropriate level of details.
5) It is not a common practice to have particular technical elements and devices presented within scientific papers. This is not common, since the available devices change their availability and technical characteristics at the market.
6) The whole paper contributes with the case study for small area in Austria. If this is the case, then this paper title should have this concept of case study emphasized.
Generally speaking, this paper does not provide any contribution, since it is based on sentences that are well-known and intuitive.
Paper could be well-formulated, if it provides description, proposed tools and methods and results (having them presented related to different areas, not only one) in adaptive integration, having concerned adaptivity as a dynamic concept based on automation.
Author Response
Thank you for your valuable feedback. We revised the paper and considered your comments in the following way:
ad 1. We added an outline of the contribution of our research project to the introduction. Our research project responds to the current practice and demands. It aims to contribute to a more sustainable practice, to establish a framework that fosters an interdisciplinary project development, that currently is mainly led by politicians and engineers, while spatial planning frameworks are not yet sufficiently developed and the expertise of disciplines such as landscape architecture is included only in rare cases. A developed procedure that seems simple and obvious does not mean that it is unnecessary to formulate it or to base it on scientific arguments. An easily understandable procedure is in this context an advantage - as it means it can be easily understood and implemented by different groups of decision-makers, including the incorporation into spatial planning guidelines.
We now referenced investigations that state the lack of consideration of unique site qualities and functional synergies of renewable energy plants in the introduction. We see the innovation of our results that the developed method provides a procedure and a toolkit for a new mainstream practice for the design of solar farms that enhance unique site qualities and provide functional synergies by considering the whole array of technical possibilities of solar modules - which currently takes place only in exceptional cases and without systematic, standardized design approach.
ad 2. We added the suggested background and related works with definitions of key terms to the introduction.
ad 3. We included a diagram for the developed methodology with qualitative input and output as Figure 8 to the paper.
ad 4 a. The investigation of automatized adaptive solutions and parametric digital designing is an interesting field. Our research project has a different focus, that is the consideration of a holistic understanding of the spatial configuration of places as well as their overall functional relevance when building solar farms, and showcasing a design approach that ensures that the design of solar power units corresponds with the unique site conditions through bespoke solutions. The research project has the goal to enrich and foster the trans-disciplinary discourse between decision makers such as political representatives, spatial planners, landscape architects, and engineers instead of digital automatization of results.
ad 4 b. Our topic relates and contributes to the smart city discourse among other topics. Our focus here is to extend the concept beyond the city, to point out the potential of smart solutions for rural decentralization and balancing the inequality between cities and the countryside.
ad 5. Any scientific paper is based on the current state of the discourse, which may be outdated a few years later. The catalogue of exemplary solar modules depicts the current state of technical development. It provides designers with a toolkit and makes them aware of already existing possibilities for designing with solar modules. For many designers as well as political representatives, with whom we discussed the research progress, this catalogue was eye-opening, because they mostly knew only the mainstream mono- or polycrystalline panels. Making aware and gaining knowledge of the different options was a pre-condition and necessary step for applying these options in the design cases during the research process. We added an explanation that the catalogue depicts the status quo to section 3.2. The nature of technical innovation and development trends that needs to be considered are discussed in section 4.4.
ad 6. The research project follows the approach of research by design, which - by its nature - investigates design case(s) and draws general realizations from the design experiment through discussion of the process with the wider discourse.
We added a statement of why the selected region was chosen for investigation. It was selected because of its diversity - both in terms of functions (open spaces of schools, business parks, town squares, different types of infrastructure, sports grounds, parks, agricultural spaces) as well as natural conditions (hilly topography, flat land, water bodies). In total, we investigated 23 different areas.
The developed method of landscape architectural design with solar modules as the main outcome of the paper is a further development of the general method of landscape architectural design by learning from the design experience in a general sense. We do not suggest applying the developed designs to other places, but developing the design case by case in a bespoke in response to the unique situation following a general procedure, which is applicable to any project dealing with the design with solar modules. To further explain this approach, we extended and re-organised sections 1. introduction and 2. materials & methods.
Reviewer 4 Report
Comments and Suggestions for Authors
Reviewer report: “Solar Landscapes: A Methodology for the Adaptive Integration of Renewable Energy Production into Cultural Landscapes”
By Chrili Car, Erwin Frohmann and Dagmar Grimm-Pretner
This is a very interesting article that highlights an issue with the integration of solar energy sources. This article really provoked me to reflect because my experience in architecture is limited to hydropower. The manuscript is clearly written, presented, and illustrated.
The introduction is concise but well-documented with thirteen important citations.
One feature of human society is anthropogenic landscapes. The environment's landscape has undergone alterations as a result ancestors' traditions of and neighboring civilizations.
The existing cultural landscapes are being profoundly transformed by outdoor solar farms. This cultural landscape represents a cultural heritage that is preserved in modern society.
The present manuscript aims at how solar energy utilization might be incorporated into these landscapes while maintaining and enhancing each location's distinctive regional identity. The research article created a conceptual design approach that starts with landscape architectural and visual analyses of sites that are currently in Styria, Austria; it also considers the spatial features of the cultural landscapes that these sites are embedded in, as well as their suitability for solar energy generation.
The analysis considers synergies with other pertinent on-site functions so that the integration of solar energy complements the features of the current landscape.
The Leibnitz district in southern Styria, Austria, serves as a reference landscape for the research study on the integration of solar thermal and photovoltaic energy in rural open areas. A diverse cultural context combined with an economic structure that places a premium on services, tourism, and agriculture allow for the testing of multiple scenarios for the integration of solar energy while accounting for synergies with different existing uses.
The method for choosing appropriate solar modules and their impact on design to create special site-specific solutions is remarkable.
The approach is innovative as it involves applying a catalog of solar modules, working with the selected sites after a thorough investigation, and exposing a five-step design process that supports a landscape-sensitive and site-specific integration of solar elements.
Value engineering and overall spatial effect assessment aim to cement the location's spatial identity while also adding locally relevant social, ecological, and economic value in addition to energy production. In order to maximize synergies in response to spatial, social, ecological, and economic requirements, the design idea is reviewed and enhanced while taking into consideration the multifunctionality of the chosen site, both now and in the future.
The article discusses the general public's "not in my backyard" attitude regarding the creation of renewable energy. While most people agree that the renewable energy sector ought to be supported, many oppose its expansion in nearby areas.
The article proposes solar systems that account for the unique characteristics of the landforms. The solar landscapes' all-encompassing and integrated design enhances the functional diversity of the landscape zones by, for instance, adding to the niche's structure and ecological mosaic while also formally adhering to the landscape's unifying qualities. Similar to an inclusive design that considers the features of the natural landscape as well as the place's ecological, social, and economic significance, the human landscape's qualities and references to the landscape's unifying elements are essential for garnering broader acceptance and ultimately leading to the creation of unique, site-specific solutions.
The research recognizes the significance of further integrating designs that use renewable energy into the frameworks of spatial planning and the necessity of incorporating technical and engineering innovations in adaptive solar modules. The method introduces landscape architecture as a key discipline in transforming the energy sector.
Since the manuscript provides information that readers of the journal will find interesting, it should be published. The outcomes of this thorough and nuanced investigation are relevant.
I commend the writers for their method of scientific refinement.
Best wishes,
The reviewer
Author Response
Thank you for your positive feedback. We are excited to hear that the research topic and outcome are of interest to you! We considered your feedback that the description of the methods can be improved: We restructured the chapter methods with a general introduction and the different applied methods as sub-chapters.
Round 2
Reviewer 1 Report
Comments and Suggestions for Authors
“ensures—case by case—that” on line 180 maybe "ensures case-by-case that"?
Comments on the Quality of English Languagefine
Author Response
Thank you for your comment, we corrected the hyphenations on line 180.
Reviewer 3 Report
Comments and Suggestions for Authors
Authors addressed all required changes to the manuscript - made necessary clarifications in additional text and added appropriate parts (such as Figure 8 with the essence of the contribution). Authors also described all changes made, according to review demands. In some cases, changes were not made, but authors supported their approach with acceptable explanations. This version is enhanced and could be accepted.
Author Response
Thank you again for your previous comments, which helped us to improve the paper, and for approving a publication now!